# Evaluating the effectiveness of speed humps and rumble strips in improving pedestrian safety in Addis Ababa, Ethiopia

Getu Segni Tulu[1]*, Robert Tama Lisinge[2], Abrham Gebre Tarekegn[1], Tigist Eshetu[1]

1 School of Civil and Environmental Engineering, College of Technology and Built Environment, Addis Ababa University, Addis Ababa, Ethiopia, 2 Technology, Innovation, Connectivity and Infrastructure Development Division, United Nations Economic Commission for Africa, Addis Ababa, Ethiopia

* getusegne@yahoo.com, getu.segni@aait.edu.et

## Abstract

Pedestrian fatalities remain a major road safety problem in Addis Ababa. To address this problem, speed humps and rumble strips were introduced on high-crash road segments in the city in 2017. As a case study that can be useful elsewhere, the aim of this research is to evaluate the effectiveness of speed hump with rumble strips in Addis Ababa. This study uses descriptive and inferential statistics to evaluate the impact of these interventions. Crash data between 2014 and 2021, excluding 2017 when the measures were implemented, is used in the analysis. The interventions led to a 25.08% reduction in operating speeds, with statistically significant speed reductions for passenger cars. Our findings on the extent to which speed humps and rumble strips lowered the speed of buses and trucks are not conclusive. Notable speed drops were observed at humps in some road segments while humps had no appreciable effect on speed in others. This underscores the need for additional measures or other approaches to reduce the operating speed of buses and trucks. The equivalent crash number for pedestrian crashes reduced by 24.45% and total crashes dropped by 51.14%. While crashes involving pedestrians crossing roads reduced, those involving pedestrians walking along travel lanes increased by 15.94%. These mixed results suggests that the effectiveness of speed humps and rumble strips depends on the type of crash being addressed and the category of vehicle involved.

## 1. Introduction

According to the World Health Organization (WHO), 1.19 million people died on the roads worldwide in 2021 [1]. In Ethiopia, traffic crashes had severe impacts between 2013 and 2022. During that period, 40,961, 64,366, and 60,771 people were killed, seriously injured, and slightly injured, respectively [2]. Road traffic crashes claim 4,097 lives in Ethiopia on average each year, amounting to around 12 deaths every day or one every two hours.

**Data availability statement:** All relevant data are within the paper and its Supporting Information files.

**Funding:** The author(s) received no specific funding for this work.

**Competing interests:** No authors have competing interests Enter: The authors have declared that no competing interests exist.

Pedestrians bear the brunt of traffic crashes, especially in cities. In 2022, urban roadways accounted for 33.5% of Ethiopia's fatal collisions, with pedestrians making up most of the victims. They constitute 80–90% of traffic deaths in Addis Ababa [3,4]. Inadequate pedestrian infrastructure, excessive speeding, drunk driving, poor street lighting, poor land-use planning, weak regulatory frameworks, and ineffective enforcement of current laws are contributing factors to the high level of pedestrian road deaths in urban areas [1,5]. Excessive speeding is a major risk factor for traffic crashes, increasing both their likelihood and severity [6]. Speed humps are among measures introduced around the world to control speed. As part of a focused effort to lower pedestrian-related crashes, Addis Ababa City Administration identified sites with high pedestrian crash rates and installed speed humps and rumble strips on those sites.

Prior studies demonstrate the effectiveness of speed-calming strategies in reducing traffic crashes. Speed humps, for example, have been shown to reduce crashes involving public transport vehicles by over 36% and all crashes by 20% [7]. According to Pan Liu [8], traverse rumble strips reduce collision frequency by 25% in the vicinity of pedestrian crosswalks.

Speed humps were initially installed in Addis Ababa in 2017 to improve road safety and curb excessive speeding [9,10]. However, to our knowledge, there are no studies on the safety performance of speed humps combined with rumble strips in the Ethiopian context. The effectiveness of the combination of speed humps and rumble strips in lowering vehicle speeds and preventing pedestrian collisions has not been assessed since their installation in selected locations in Addis Ababa. Furthermore, the impact of speed humps and rumble strips on the speed levels of different vehicle types such as passenger cars, buses, and trucks has not been investigated. The objective of this study is therefore to investigate the extent to which speed humps and rumble strips reduce vehicle speeds and pedestrian collisions across various vehicle categories in Addis Ababa.

The study compares pedestrian crash data before and after the installation of speed humps and rumble strips to investigate the effectiveness of these measures. Its null hypothesis is that there is no discernible difference between the number of pedestrian crashes before and after speed humps are installed. The alternative hypothesis is that there is notable change in pedestrian crash rate after the intervention. In this regard, most vehicles in Addis Ababa are old their reaction to speed-calming measures may be different from that of newer vehicles, which could have an impact on the severity of crashes. This underscores the importance of undertaking context-specific studies, such as the present study, on the impact of speed-calming measures.

The characteristics of vehicles in a city, their operating speed, and other factors such as road geometry, and ambient conditions determine the effect of speed humps and rumble strips on crashes involving pedestrians. In this regards, most vehicles in Addis Ababa are old and their reaction to speed calming measures may be different from that newer vehicles, which could have an impact on the severity of crashes [11]. This underscores the importance of undertaking context-specific studies, such as the present study, on the impact of speed-calming measures.

This study contributes to the literature by assessing the safety performance of speed humps in low-and middle-income countries (LMICs), specifically in Addis Ababa in Ethiopia, with a particular focus on 30 km/h speed humps combined with rumble strips. The study evaluates the effectiveness of this combination of measures in reducing operating speeds across different vehicle categories and the contribution of the combined measures to crash reduction on selected road segments in the city.

The next section details the methodology of this study, including data collection and analytical approaches. The results section presents descriptive and inferential findings, shedding light on the impact of speed calming measures on road safety in Addis Ababa.

## 2. Materials and methods

The data collection procedure, which included obtaining both primary and secondary data, is described in this section. Observations from spot speed surveys that measured operating speeds by vehicle type where part of the primary data collection. Secondary data consisted of crash records, including records on fatal, serious, and minor injuries, with a focus on pedestrian crashes.

### 2.1. Spot speed survey

Laser Radar technology, provided by the Bloomberg Road Safety Initiative, was used because of its high precision and advanced capabilities. This technology utilizes a velocity speed gun, capable of measuring vehicle speeds ranging from 0 to 320 km/h, with a maximum measurement distance of 1,200 meters. To ensure the accuracy of the speed measurements, the radar path had to closely align with the route of the vehicle being monitored. When the measurement was taken in a straight line, the recorded speed values were accurate to within ±2 km/hr.

Two locations were selected for data gathering for the speed survey. The first location was prior to the speed hump and the second was right at the speed hump. The study sought to determine how well speed calming strategies reduced vehicle speeds and in turn, the risk of crashes by comparing the speeds observed at these intervals.

Speed data for various categories of vehicles was methodologically gathered and precisely documented using a common survey form. To minimize traffic disturbance, the surveys where carried out off-peak hours, namely from 10:00 am to noon and from 2:00pm to 4:00pm. To guarantee accurate results, the literature recommends that at least 50 vehicles, ideally 100 vehicles, should be sampled at each location [8]. Following this guidelines, 100 vehicles were observed at each site both before and after speed humps.

The vehicles were divided into three categories, namely car, buses, and trucks to comprehend the variations in speed patterns and the corresponding crash probabilities. This classification enabled a comprehensive examination of the effects of combing speed and rumble strips. It also enabled the varying impacts of speed calming measures across vehicle categories to be determined.

### 2.2. Crash data

In line with the WHO injury severity categorizations in Ethiopia: fatal, serious injury, minor injury and property damage only [3]. A 30-day time frame is used to gauge the severity of injuries from a crash. If a crash victim passes away within 30 days after a crashes, the crash is deemed fatal. Mino or slight injury crashes are those in which at least one person needs hospital or outpatient care for less than twenty four hours, where serious injury crashes involve at least one person being hospitalized for twenty four hours or more.

The Addis Ababa Commission's crash data recorded provided the fatality crash data used in this investigation. The city is divided into 11 sub cities. In this regards, information on property damage, minor injuries, and serious injuries were gathered from the respected police departments of the subsidies, with an emphasis on pedestrian injury crashes. Notably, crash data in Ethiopia, like in many other less developed countries, may be impacted by under reporting [12].

Information was gathered from eight sub cities with speed humps are found, and these include Yeka, Lideta, Arada, Gulele, Kolfe, Keraniyo, and Nifasilk Lafto. Crashes resulting in fatalities or significant injuries are less likely to be underreported because victims or their families usually seek police records for insurance and legal purposes.

The collected crash data, including fatal, serious, and slight injuries, were converted into equivalent crash numbers (ECNs) to account for both crash frequency and injury severity at each site. A weight of 6 was assigned to fatal injuries, 4 to serious injuries, and 2 to slight injuries. The slight injury data was excluded with the view to focusing on prevention of fatalities and serious injuries.

The study was conducted between 2014 and 2021, excluding 2017 when the speed humps and rumble stips were installed. The three years prior to implementation of the measures from July 2014 to June2017, constituted the baseline for the analysis. The time frame following the intervention was from July 2018 to June 2021. This long period was selected to reduce the possibility of distortions from abnormalities in any one year and improve the trustworthiness of the results. By using this method, the study sought to guarantee that the findings were solid and representative of long term patterns rather than transient fluctuations.

## 2.3. Method of data analysis

Researchers worldwide have used a variety of statistical techniques to assess the safety performance of speed humps. For example, a study in South Africa examined vehicle-pedestrian collisions on roads with speed humps using observational and interrupted time series (ITS) before-and-after study methods [13]. The results showed that installing speed humps resulted in a statistically significant decrease in crashes.

In a similar vein, a Malaysian study evaluated speed drops close to speed humps. Using t-tests, the data showed that although vehicle speeds stayed high as they approached the humps, they dramatically dropped by 46% for round-top humps and 52% for flat-top humps [8].

Multivariate conditional logistic regression technique was used in an Oakland, California, study to assess how well speed humps reduce pedestrian collisions [14]. Applying ANOVA, this study examined variations in operating speed and pedestrian crash rates. Using pedestrian crash data, paired sample t-tests were also employed to examine the difference between the means of two matched groups.

## 2.4. Permits obtained

Addis Ababa University officially requested permission for the collection of data essential to our study from the city's traffic management agency. The request was done through a formal letter that defined the scope of the study and the data collection requirements. This made it easier to obtain crash and speed data for the period comprising of three years prior to the intervention (installation of the speed hump and rumble strips) and three years after the intervention. Additionally, permission to perform fieldwork was obtained from the Addis Ababa City Administration Police Commission. All study activities were carried out strictly in accordance with the rules outlined in these permissions.

## 3. Results

### 3.1. Description of data

Thirteen road segments were chosen for this study, most of which had histories of high level of crashes. The road geometry, ambient conditions, and operational parameters of these road segments varied in length from 0.4km to 2.8 km and their average length is 1.26km. They are characterised by steep grades, heavy motorized and pedestrian traffic, and a dense population living beside the roadways.

Additionally, the fleet of vehicles in these segments, including public transport and heavy vehicles are noticeable older than elsewhere in Addis Ababa [11,15]. Most of these road segments have several bridge and sudden change in geometry, including combinations of horizontal and vertical curves. Drivers find it challenging to navigate these intricate road

situations. According to 2017 study, speeding and inadequate lighting result in frequent sever injury crashes, especially at night, on the same of the selected road segments [3].

Three different kinds of speed humps with rumble strips were installed in Addis Ababa. They were made to accommodate cars traveling at 30 km/h, 40 km/h, and 50 km/h. In terms of specifications and geometric design, these speed humps complied with international norms. The 30 km/h round-top speed humps installed at the 13 locations indicated in Table 1 were the primary focus of the current investigation. The chosen locations were crucial targets for modification since they had poor geometric designs and a high crash rate. The operational, site-specific, and environmental features of the selected road sections presented in Table 1 provided context for the analysis in this article. Summary of the data are presented in Table 2.

### 3.2. Analysis of speed survey data

To assess how well speed humps and rumble strips reduce operating speeds, the study used a spot speed survey to record travel speeds across all road segments included in the analysis. This survey played crucial role in obtaining information and determining how the interventions affect speeds. The 85th percentile spot speeds, or operating speeds, that

**Table 1. Environmental and operational features of road segments.**

| No. | Road segments | Length (km) | Description of the road segment |
|---|---|---|---|
| 1 | Bethel Roundabout - Keranio Roundabout | 1.8 | • Very steep road section<br>• Has two unsignalised intersections (T and Y)<br>• Located in high commercial and residential areas<br>• High vehicular and pedestrian flows |
| 2 | Anfo Roundabout - Keranio Roundabout | 2.8 | • Located at peripheral part of the city<br>• Steep road section |
| 3 | Mendida intersection - Tor hayloch | 2.4 | • Substandard curves; mis-phasing horizontal and vertical curves with bridge in both directions<br>• Steep road section |
| 4 | Lideta interchange - African Union Junction (LidetaTsebel) | 0.9 | • Steep section |
| 5 | Mekanisa Abo Roundabout to Mekanisa bridge | 1.0 | • Very steep section<br>• High volume traffic |
| 6 | BisrateGebriel - Mekanisa Abo Roundabout | 1.7 | • Church and schools for students who have listening problems located in vicinity of road section |
| 7 | Lideta Park - Lideta Condominium-Lideta interchange | 0.6 | • Densely populated area due to the presence of low-cost houses |
| 8 | Semen hotel - Afinchober | 0.9 | • Very steep road section<br>• High traffic volume<br>• Signalized intersection |
| 9 | Balderas condominium - Aware Roundabout | 1.0 | • Steep gradient<br>• High traffic volume |
| 10 | Parliament signalized intersection - Aware Roundabout | 1.4 | • Very steep section with street venders<br>• High traffic volume |
| 11 | KokebeTsibah secondary School - Kebena Roundabout | 0.4 | • Steep section, followed by roundabout |
| 12 | Minilik II hospital – Kebena Roundabout | 0.8 | • Very steep section, followed by roundabout<br>• Mis-phasing horizontal and vertical curves with bridge in both directions |
| 13 | 6 kilo RA - Afinchober | 0.7 | • Very steep section, followed by roundabout<br>• Combination of horizontal and vertical curves with bridge and roundabout in both directions<br>• School areas (Addis Ababa University) |
| | Total | 16.4 | |

**Table 2. Summary of data.**

| No. | Descriptions | No. of total observation | Minimum | Average | Maximum | Range |
|---|---|---|---|---|---|---|
| 1 | Road segment in km | 13 | 0.4 | 1.26 | 2.8 | 2.4 |
| 2 | No. of speed humps per road segment | 56 | 2 | 4.307 | 12 | 10 |
| 3 | Speed hump density (no. of speed hump/km) | | 2 | 3.4 | 5 | 3 |
| 4 | No. of rumble strips considered per site | 81 | 1 | 6.23 | 17 | 16 |
| 5 | Spot speed for passenger cars | Before the speed hump | 16.7 | 27.12 | 42.4 | 25.7 |
| | | At speed hump | 15.8 | 20.41 | 34.8 | 19 |
| 6 | Spot speed for buses | Before speed hump | 17.7 | 25.32 | 40.80 | 23.1 |
| | | At speed hump | 13.2 | 18.74 | 34.60 | 21.4 |
| 7 | Spot speed for Trucks | Before speed hump | 14.9 | 23.52 | 34.4 | 19.5 |
| | | At speed hump | 11.00 | 17.41 | 28.7 | 17.7 |
| 6 | Pedestrian fatal and serious injury counts before speed hump installation on selected sites per three years period | 235 | 4 | 18.1 | 31 | 27 |
| 7 | Pedestrian fatal and injury counts after speed hump installation on selected sites per three years period | 143 | 3 | 11 | 20 | 17 |

**Table 3. Summary of 85th percentile spot speed.**

| Road segment | Approach speed Before the speed hump | At the speed hump | Average reduction in 85th percentile spot speed | % reduction |
|---|---|---|---|---|
| Bethel Roundabout - Keranio Roundabout | 23 | 23 | 0 | 0.00 |
| Anfo Roundabout - Keranio Roundabout | 20 | 18 | 2 | 10.00 |
| Mendida intersection - Tor hayloch | 45 | 39 | 6 | 13.33 |
| Lideta interchange - African Union Junction (LidetaTsebel) | 32 | 23 | 9 | 28.13 |
| Mekanisa Abo Roundabout -Mekanisa bridge | 31 | 21 | 10 | 32.26 |
| BisrateGebriel-Mekanisa Abo Roundabout | 39 | 29 | 10 | 25.64 |
| Lideta Park - Lideta Condominium-Lideta interchange | 26 | 28 | -2 | -7.69 |
| Semen hotel signalized intersection - Afinchober | 27 | 17 | 10 | 37.04 |
| Balderas condominium to -Aware Roundabout | 27 | 17 | 10 | 37.04 |
| Parliament signalized intersection - Aware Roundabout | 28 | 18 | 10 | 35.71 |
| KokebeTsibah secondary - School Kebena Roundabout | 35 | 20 | 15 | 42.86 |
| Minilik II hospital - Kebena Roundabout | 30 | 21 | 9 | 30.00 |
| 6 kilo RA - Afinchober | 36 | 21 | 15 | 41.67 |
| **85th percentile average speed** | **30.69** | **22.69** | **8** | **25.08%** |

were recorded prior to and at the speed hump locations are shown in Table 3. Significant differences in spot speed were found in the data, impacted by contextual factors.

During the day, observations showed slower speeds, most likely because of high traffic level and the presence of vulnerable road users, which inherently make drivers slow down. Conversely, lower pedestrian and traffic level seemed to motivate faster speeds at night since drivers have fewer outside constraints under these conditions. The risk to pedestrians posed by excessive speeding at night is exacerbated by decreased visibility and copounded during weekends by alcohol consumption.

While the average speed at the speed humps was between 17 and 29km/hr, there was a noteworthy outlier, with a record speed of 39km/hr (Mendida Intersection-Tor Hayloch road segment). Crucially, the average speeds at humps were

always below the 30km/hr which corresponds to safe operating conditions. Across all the selected road segments, the study found an average speed decrease of 25.08% at humps compared to speed before the humps, which was statistically significant. These results highlight the extent to which speed humps and rumble strips reduce travel speeds and improve pedestrian safety, especially during the day.

The extent of the difference in vehicle speed before and at a hump varied across the road segments studied. For example, the 85th percentile speed was the same before and at the speed humps on the road segment between Lideta Park-Lideta Condominium and Lideta Interchange, speeds rose close to and at the humps (see Table 3)

Varying results were observed for trucks and buses across road segments. Notable speed drops were observed at humps in some road segments while the humps had no appreciable effect on speed in others. This implies that the speed humps' success in lowering the speeds of truck and buses is not conclusive. This result might be explained by the higher vertical clearance from the road surface of large vehicles compared to passenger cars. It suggests that speed humps are less effective in reducing speed for trucks and buses, underscoring the need for additional or other approaches to deal with their operating speeds.

Table 4 summarize the ANOVA results on variations in traffic speeds prior to and following the installation of speed humps and rumble strips. To evaluate the effect of the interventions on various vehicle types, the study divided vehicles into three groups categories: passenger cars, buses, and trucks.

With a p-value of less than 5% for almost all the road segments, the data shows that the interventions considerably decreased vehicle speeds for passenger automobiles. However, data from Bethel Roundabout to Keranio Roundabout road segment was not statistically significant. With nearby stores, places of worship and mosques, this road segment is a major thoroughfare characterized by heavy pedestrian traffic during the day, which may have contributed to the observed low speed levels even prior to the intervention. In contrast, higher speed levels are observed at night because of motorized and pedestrian traffic on the road, which could lessen the impact of the interventions on speed. The findings were significant for business in most road segments (statistically significant decrease in speed). The few exceptions include the Bethel Roundabout to Keranio Roundabout, Anfo Roundabout to Keranio Roundabout, and Lideta Interchange to Africa Union Headquarter Junction road segments that did not exhibit statistically significant decreases in speed following the interventions. This implies that bus speed in these sections were not significantly affected by the combination of speed humps and rumble strips. Speed studies for buses were not possible for two road segments, namely: Balderas condominium to Aware Roundabout, and Bisrate Gebriel to Mekanisa Abo Roundabout, due to a lack of data, which limited the findings for these segments.

The truck analysis revealed that speed decreases were statistically significant on seven road segments with p-values below 5%. However, the lack of statistical significance in speed reductions on six other segments, with p-values above 5%, indicates that the combination of speed humps and rumble strips was ineffective in lowering truck speeds on those road segments. The results show that although installing speed humps with rumble strips successfully lowered passenger car and lesser degree bus and truck speeds, the effect varied across road segments.

It appears that factors, such land use, traffic mix, and time of day, affect the effectiveness of speed humps and rumble strips. More research and customized solutions are required to deal with situations where speed humps and rumble strips fail to reduce excessive speeding.

## 3.3. Analysis of traffic crash data

Data on injury severity was gathered for three years prior to and three years following the installation of speed humps and rumble strips. Equivalent Crash Numbers (ECNs) were calculated using previously assigned weights for various crash severity levels to assess the effectiveness of these interventions. The average ECN for all the road segments for the three years pior to the interventions was calculated. The average ECN for the post-crash period was also calculated.

**Table 4. Summary of ANOVA results for traffic speed.**

| | Speed humps Locations by road segment | F- Value at the 0.05 level | | | | F- Value at the 0.05 level | p-value>/ z/ |
|---|---|---|---|---|---|---|---|
| | | Before speed hump | | After speed hump | | | |
| | | Mean | Std | Mean | Std | | |
| **Summary of ANOVA results for Car** | Balderas | 22.442 | 4.573 | 15.043 | 2.380 | 189.456 | **0.000*** |
| | Bethel | 20.949 | 5.784 | 20.506 | 3.194 | 0.355 | 0.552 |
| | Anfo | 16.951 | 15.790 | 3.798 | 3.193 | 4.432 | |
| | Mendida | 41.436 | 4.842 | 36.864 | 5.264 | 27.775 | **0.037**** |
| | Lideta Interchange | 27.714 | 4.632 | 19.357 | 4.297 | 122.474 | **0.000*** |
| | Mekanisa-Abo | 25.311 | 6.595 | 18.909 | 5.365 | 28.664 | **0.000*** |
| | BisrateGebriel | 34.253 | 5.587 | 24.301 | 4.856 | 146.826 | |
| | Lideta Interchange Condominium | 20.584 | 5.342 | 25.385 | 5.149 | 25.796 | **0.000*** |
| | Semen Hotel | 22.800 | 4.335 | 15.267 | 2.811 | 162.415 | **0.000*** |
| | Aware Roundabout | 20.473 | 6.272 | 14.769 | 3.432 | 58.202 | **0.000*** |
| | Kokebe-Tsibah | 30.409 | 7.591 | 18.143 | 4.534 | 112.160 | **0.000*** |
| | MinilikKebena | 26.730 | 4.900 | 18.238 | 4.134 | 110.557 | **0.000*** |
| | 6 killoAfinchober | 32.683 | 5.177 | 18.726 | 3.398 | 316.433 | **0.000*** |
| **Summary of ANOVA results for Bus** | Balderas | | | | | | **No Enough Data** |
| | Bethel | 19.5 | 10.607 | 20 | 5.138 | 0.0092 | **0.927** |
| | Anfo | 18.4 | 5.814 | 11.5 | 2.121 | 2.434 | **0.179** |
| | Mendida | 36.7 | 31.875 | 4.398 | 4.969 | 4.771 | **0.044**** |
| | Lideta Interchange | 25.667 | 17.75 | 5.680 | 4.113 | 5.674 | **0.044**** |
| | Mekanisa-Abo | 24.933 | 16.385 | 3.575 | 3.228 | 43.527 | **0.000*** |
| | BisrateGebriel | | | | | | **No Enough Data** |
| | Lideta Interchange Condominium | 18.875 | 4.086 | 17.833 | 2.787 | 0.287 | **0.602** |
| | Semen Hotel | 24.2 | 14 | 5.069 | 1 | 19.483 | **0.0022**** |
| | Aware Roundabout | 0 | -- | 10.667 | 2.517 | 13.47368 | **0.067*** |
| | Kokebe-Tsibah | 30.667 | 15.667 | 2.517 | 3.172 | 56.919 | **0.000*** |
| | MinilikKebena | 24.111 | 13.4 | 6.133 | 2.591 | 25.571 | **0.000*** |
| | 6 killoAfinchober | 25.857 | 14.222 | 8.153 | 2.991 | 15.863 | **0.0014**** |
| **Summary of ANOVA results for Trucks** | Balderas | 15.25 | 4.652 | 11.25 | 4.031 | 2.13067 | **0.175** |
| | Bethel | 18 | 5.895 | 17.188 | 4.549 | 0.148 | **0.703** |
| | Anfo | 19.286 | 9.725 | 15.882 | 3.5511 | 1.642 | **0.213** |
| | Mendida | 33.833 | 5.287 | 29.818 | 5.845 | 2.992 | **0.098*** |
| | Lideta Interchange | 27.541 | 4.180 | 17.36 | 6.473 | 42.381 | **0.000*** |
| | Mekanisa-Abo | 27.605 | 4.353 | 17.115 | 3.559 | 103.484 | **0.000*** |
| | BisrateGebriel | 24.75 | 5.280 | 23.5 | 3.949 | 0.618 | **0.437** |
| | Lideta Interchange Condominium | 20.533 | 5.330 | 21.093 | 4.129 | 0.158 | **0.695** |
| | Semen Hotel | 23.285 | 3.244 | 14.2 | 2.706 | 78.804 | **0.000*** |
| | Aware Roundabout | 17.285 | 4.461 | 9.667 | 2.422 | 13.867 | **0.003*** |
| | Kokebe-Tsibah | 26.067 | 5.0918 | 16.03226 | 3.381 | 82.725 | **0.000*** |
| | MinilikKebena | 22.714 | 4.108 | 14.333 | 3.082 | 31.490 | **0.000*** |
| | 6 killoAfinchober | 26.833 | 5.687 | 15.965 | 2.958 | 83.906 | **0.000*** |

Note ***, **, * significant at 1%, 5%, 10% level respectively.

Table 5 shows the ECNs for all pedestrian crashes before and after implementation of speed humps and rumble strips. Table 6 shows the ECNs for crashes related to pedestrians crossing the road before and after implementation of speed humps and rumble strips. Table 7 shows the ECNs for crashes involving pedestrians walking along travel lanes before and after implementation of speed humps and rumble strips.

The ECN for all pedestrian crashes dropped by 24.45% when speed hums and rumble strips were installed, indicating a notable decrease in injuries (Table 5). This suggests that the frequency of pedestrian injury crashes was successfully reduced by the intervention.

Likewise, there was a 51.4% decrease in ECNs for pedestrian crossing crashes (Table 6) However, the ECN for crashes involving pedestrians walking on travel lanes increased by 15.94% (Table 7), suggesting a potential unintended consequence of the intervention in this specific context.

**Table 5. Comparison of Equivalent Crash Numbers of pedestrian crashes before and after implementation of speed humps with rumble strips.**

| Road Name | Pedestrian ECNs Before | Pedestrian ECNs After | Change in ECNs | % reduction |
|---|---|---|---|---|
| Bethel RA - Keranyo RA | 34 | 36 | 2 | 5.88 |
| Anfo RA - Keranyo RA | 74 | 72 | -2 | -2.70 |
| Mendida intersection - Tor hayloch | 30 | 36 | 6 | 20.00 |
| Lideta interchange - AU Junction (LidetaTsebel) | 126 | 56 | -70 | -55.56 |
| Mekanisa Abo RA - Mekanisa bridge | 58 | 54 | -4 | -6.90 |
| BisrateGebriel - Mekanisa Abo RA | 56 | 32 | -24 | -42.86 |
| Lideta Condominium - Lideta interchange | 76 | 76 | 0 | 0.00 |
| Semen hotel signalized intersection- Afinchober | 60 | 52 | -8 | -13.33 |
| Balderas condominium - Aware RA | 18 | 24 | 6 | 33.33 |
| Parlama signalized intersection-Aware RA | 72 | 44 | -28 | -38.89 |
| KokebeTsibah secondary school - Kebena RA | 26 | 8 | -18 | -69.23 |
| Minilik II hospital - Kebena RA | 34 | 32 | -2 | -5.88 |
| 6 kilo RA - Afinchober | 48 | 16 | -32 | -66.67 |
| average | 54.8 | 41.4 | -13.4 | -24.45 |

**Table 6. Comparison of Equivalent Crash Numbers of crossing-related pedestrian crashes before and after implementation of speed humps with rumble strips.**

| Road Name | Crossing Pedestrian ECNs Before | Crossing Pedestrian ECNs After | Change | % reduction |
|---|---|---|---|---|
| Bethel RA - Keranyo RA | 22 | 6 | -16 | -72.73 |
| Anfo RA - Keranyo RA | 46 | 24 | -22 | -47.83 |
| Mendida intersection - Tor hayloch | 30 | 6 | -24 | -80 |
| Lideta interchange - AU Junction (LidetaTsebel) | 84 | 40 | -44 | -52.38 |
| Mekanisa Abo RA - Mekanisa bridge | 34 | 14 | -20 | -58.82 |
| BisrateGebriel - Mekanisa Abo RA | 38 | 10 | -28 | -73.68 |
| Lideta Condominium - Lideta interchange | 62 | 14 | -48 | -77.42 |
| Semen hotel signalized intersection - Afinchober | 38 | 36 | -2 | -5.26 |
| Balderas condominium - Aware RA | – | 10 | 10 | |
| Parlama signalized intersection - Aware RA | 52 | 24 | -28 | -53.86 |
| KokebeTsibah secondary school - Kebena RA | 16 | 8 | -8 | -50 |
| Minilik II hospital - Kebena RA | 28 | 16 | -12 | -42.86 |
| 6 kilo RA - Afinchober | 16 | 8 | -8 | -50 |
| Average | | | | -51.14 |

T-tests were used to assess the null hypothesis that there is no discernible difference between the number of pedestrian crashes before and after the speed humps and rumble strips were installed. First the analysis was performed for total crashes (all categories of crashes). The results shows that the mean number crashes before and after the intervention period differed significantly. Specifically, a p-value of less than 1% indicates that the change in Equivalent Crash Numbers (ECNs) is statistically significant (Table 8). The fact that installing speed humps and rumble strips is linked to a significant decrease in the number of collisions shows that they are effective in enhancing traffic safety.

Second, pedestrian collisions that happened three years before and three years after the installation of speed humps and rumble strips were analysed. The t-test shows a statistically significant reduction in pedestrian crashes, with a p-value of less than 5% (Table 8). Pedestrian collisions are a significant safety concern in Addis Ababa, accounting for 80–90% of all annual road fatalities in the city, as mentioned in Section 1 [3]. Thus, installing speed humps and rumble strips are priority measures to save lives in the city.

Third, pedestrian injuries and fatalities at road crossing were analysed. With a p-value of 1%, the analysis shows a strong correlation between the installation of speed humps and rumble strips crashes involving pedestrians crossing roads (Table 8). This findings demonstrates that a combination of speed humps and rumble strips improves pedestrian safety, particularly by lowering collisions at intersections. Other researches produced similar results. For instance,

**Table 7. Comparison of crashes involving pedestrians walking on travel lanes before and after implementation of speed humps with rumble strips.**

| Road Name | Walking Pedestrian ECNs Before | Walking Pedestrian ECNs After | Change | % reduction |
|---|---|---|---|---|
| Bethel RA to Keranyo RA | 6 | 2 | -4 | -66.67 |
| Anfo RA to Keranyo RA | 4 | 16 | 12 | 300 |
| Mendida intersection to Tor hayloch | 0 | 6 | 6 | – |
| Lideta interchange to AU Junction (LidetaTsebel) | 22 | 8 | -14 | -63.64 |
| Mekanisa Abo RA to Mekanisa bridge | 16 | 6 | -10 | -62.5 |
| BisrateGebriel to Mekanisa Abo RA | 12 | 10 | -2 | -16.67 |
| Lideta Condominium to Lideta interchange | 10 | 20 | 10 | 100 |
| Semen hotel signalized intersection to Afinchober | 4 | 8 | 4 | 100 |
| Balderas condominium to Aware RA | 0 | 8 | 8 | – |
| Parlama signalized intersection to Aware RA | 16 | 8 | -8 | -50 |
| KokebeTsibah secondary school to Kebena RA | 2 | 0 | -2 | -100 |
| Minilik II hospital to Kebena RA | 6 | 4 | -2 | -33.33 |
| 6 kilo RA to Afinchober | 4 | 8 | 4 | 100 |
| Average | | | | 15.94% |

**Table 8. T-test results before and after the implementation of speed hump and rumble strips using equivalent crash numbers.**

| Pairs (before and after) | Change in Mean ECNs | Std. Deviation | Std. Error Mean | P-value | 95% Confidence Interval of the Difference | |
|---|---|---|---|---|---|---|
| Change in total crashes | 28.769 | 25.133 | 6.971 | **0.001***** | 13.581 | 43.957 |
| Change in total pedestrian crashes | 13.385 | 21.298 | 5.907 | **0.043**** | 0.515 | 26.255 |
| Change in crossing Pedestrian crashes on roadway | 19.231 | 16.053 | 4.452 | **0.001***** | 9.530 | 28.931 |
| Change in pedestrian crashes walking on travel lanes | -0.154 | 7.978 | 2.213 | 0.946 | -4.975 | 4.667 |

Note ***, **, *significant at 1%,5%,10% level respectively.

undesignated midblock crossings were found to be major predictors of pedestrian injury collisions in Michigan, USA [16]. A study undertaken in Canada also showed that uncontrolled midblock crossing had greater crash rates than managed intersection [17].

Finally, the present study analyzed crashes involving pedestrians walking along travel lanes, both with and against the flow of traffic. The results indicate that the difference in mean crash numbers before and after the installation of speed humps and rumble strips was statistically insignificant. This suggests that the frequency of such pedestrian crashes remained relatively consistent regardless of the intervention. It supports the null hypothesis that there is no discernible difference between the number of pedestrian collisions before and after speed humps and rumble strips were installed in the context of pedestrians using traffic lanes as walkways.

## 4.  Discussions

The impact of a combination of speed humps and rumble strips on pedestrian collision rates and the severity of pedestrian injuries has been investigated. Evaluating vehicle operating speeds close to, and at speed hump locations was a crucial part of this investigation. The mean operating speed or the 85th percentile speed remained below 30km/hr after these traffic calming measures were implemented. This study shows that installing speed humps and rumbles strips significant.

Previous research showed that the likelihood of a fatal pedestrian crash is less than 10% if the vehicle involved travels at a speed of 30km/hr or less, hence lowering speed is essential to improving pedestrian safety. A combination of speed humps and rumble strips lowers the probability of pedestrian crashes and severity of pedestrian injuries by maintaining operating speeds below the 30km/hr limit. Installing speed humps and rumble strips is therefore important to improve road safety in cities with considerable pedestrian traffic.

This study also revealed that a few cars travelled faster than the posted speed limit of 30 km/h at treated places, highlighting the problem of non-compliance with speed limits, and underscoring the need to enforce compliance. This could include more police patrols or the use of automated systems (like speed cameras). Campaigns to raise public awareness on the importance of complying with speed limits could also be undertaken. These supplementary actions could enhance the effectiveness of speed humps and rumble strips in creating safer pedestrian environments and reducing overall crash severity.

Three different vehicle categories were used in the study: passenger cars, buses, and cargo vehicles (such as trucks). ANOVA testing showed statistically significant differences in average speeds in the passenger car group before and after the installation of speed humps and rumble strips. The analysis demonstrated that the average speed of passenger automobiles considerably decreased after the interventions. These results illustrate that speed humps are effective in slowing down light vehicles, particularly when paired with rumble strips.

However, this speed reduction comes with unintended consequences. Reduced vehicle speeds can contribute to air pollution, which is a major problem in metropolitan areas. Research in the US shows that cars that move at about 80 km/h produce the lowest level of emissions (Suyabodha, 2017) [18]. Lower speeds result in a significant increase in emissions, which exacerbates urban air quality issues. Additionally, Haroon [19] points out the harmfulness, especially in residential areas, of noise pollution from cars going over speed humps. Municipal governments are faced with the challenge of optimising the benefits of speed humps and rumble strips while addressing their negative consequences.

The analysis of changes in speed levels before and after the installation of speed humps and rumble strips produced inconsistent results for trucks and buses. Some road segments experienced statistically significant decreases in the speed of these categories of vehicle while others did not. The reason for this discrepancy is that buses and trucks have greater vertical clearance compared to passenger cars, which lessens the effect of speed hump-induced vertical deflection. This suggests that speed humps may be less effective in reducing the speed of heavy vehicles than light ones.

This study further explored the relationship between speed humps and rumble strips and crash outcomes. The analysis shows that all injury crashes (pedestrian and non-pedestrian injury crashes), total pedestrian injury crashes, and

 

pedestrian injuries at crossings significantly declined after the interventions. These findings highlight the extent to which speed humps and rumble strips can reduce crash rates and the severity of pedestrian crash outcomes.

However, the reduction of crashes involving pedestrians walking along travel lanes was found to be statistically insignificant. This suggests that speed humps and rumble strips are less effective in preventing collisions when pedestrians walk alongside or against cars on travel lanes. It highlights the need for additional measures, such as increased lane separation or designated pedestrian pathways, to address this specific type of crash.

Overall, even though speed humps and rumble strips generally increase pedestrian safety and lower crash rates, their installation needs to be supported by plans to address related issues like higher emissions and noise pollution, particularly in crowded cities.

## 5. Limitation of the study

The underlying premise of this study is that external conditions stayed unchanged during the study period. In particular, the road segments that were observed between 2014 and 2017 and 2018 and 2021 were not rebuilt. Furthermore, changes that could have affected the finding of the study were not made to traffic laws throughout this period. The findings are unaffected by the publication of a new traffic law in August 2024, which is notable but happened after the study period. A possible drawback of this study is driver behaviour, which may change over time.

## 6. Conclusions

This study has assessed the effectiveness of a combination of speed humps and rumble strips in improving pedestrian safety in Addis Ababa. We evaluated the impact of these measures across a range of road segments and found that the 85th percentile speed stayed much below the 30 km/h speed limit, with the mean 85th percentile speed falling by 25.08%. This study demonstrates the effectiveness of speed humps in reducing vehicle speeds, especially for passenger cars. Reduction of average passenger car speed was found to be statistically significant. By lowering the speed of passenger cars, the combination of speed humps and rumble strips reduces the frequency and severity of crashes involving pedestrians and this category of vehicles, thereby improving road safety.

However, installation of speed humps and rumble strips had less impact on the speed of buses and trucks. Our analysis reveals conflicting results for this category of vehicles. Their average speeds reduced significantly in some road segments but not in others. The interventions seem to have limited ability to reduce the speed of bigger vehicles, such trucks, and buses.

This study highlights some negative effects of speed humps. For instance, reducing the speed of vehicles increases their emissions, contributing to air pollution in urban areas. Noise pollution from cars going over speed humps and rumble strips may also have a detrimental impact on the quality of life in local communities.

Even though speed bumps and rumble strips are useful tools to reduce speed and improve road safety, their installation should be accompanied by plans to address related environmental issues. To guarantee a balanced approach to urban traffic management, the city government should prioritise actions that address noise pollution in residential areas and lower emissions from slow-moving vehicles.

## Supporting information

**S1 Raw Data. Speed.**
(ZIP)

## Acknowledgments

The views expressed in this article are those of the authors and do not necessarily reflect the views of the United Nations Economic Commission for Africa or other associated parties.

## Author contributions

**Conceptualization:** Getu Segni Tulu, Abrham Gebre Tarekegn.

**Data curation:** Getu Segni Tulu, Tigist Eshetu.

**Formal analysis:** Getu Segni Tulu, Abrham Gebre Tarekegn, Tigist Eshetu.

**Investigation:** Getu Segni Tulu, Abrham Gebre Tarekegn.

**Methodology:** Getu Segni Tulu, Tigist Eshetu.

**Software:** Getu Segni Tulu.

**Supervision:** Getu Segni Tulu.

**Visualization:** Robert Tama Lisinge.

**Writing – original draft:** Getu Segni Tulu.

**Writing – review & editing:** Getu Segni Tulu, Robert Tama Lisinge.

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
