## [Decision Letter · Decision Letter 0]

2 Oct 2024

PONE-D-24-38618Evaluation of the Effectiveness of Speed humps with Rumble Strips on Pedestrian Safety in Addis Ababa, EthiopiaPLOS ONE

Dear Dr. Tulu,

Thank you for submitting your manuscript to PLOS ONE. After careful consideration, we feel that it has merit but does not fully meet PLOS ONE’s publication criteria as it currently stands. Therefore, we invite you to submit a revised version of the manuscript that addresses the points raised during the review process.

We look forward to receiving your revised manuscript.

Kind regards,

Mitiku Badasa Moisa

Academic Editor

PLOS ONE

Journal Requirements: When submitting your revision, we need you to address these additional requirements. 1. Please ensure that your manuscript meets PLOS ONE's style requirements, including those for file naming. The PLOS ONE style templates can be found at https://journals.plos.org/plosone/s/file?id=wjVg/PLOSOne_formatting_sample_main_body.pdf and https://journals.plos.org/plosone/s/file?id=ba62/PLOSOne_formatting_sample_title_authors_affiliations.pdf 2. In your Methods section, please provide additional information regarding the permits you obtained for the work. Please ensure you have included the full name of the authority that approved the field site access and, if no permits were required, a brief statement explaining why.

**Additional Editor Comments:**

Evaluation of the Effectiveness of Speed humps with Rumble Strips on Pedestrian Safety in Addis Ababa, Ethiopia

PONE-D-24-38618

Academic editor comment

Your manuscript entitled " Evaluation of the Effectiveness of Speed humps with Rumble Strips on Pedestrian Safety in Addis Ababa, Ethiopia", which you submitted to PLOS ONE, has now been reviewed.

The reviews, included at the bottom of the letter, indicate that your manuscript could be suitable for publication following revision. The comments of the reviewers are critical and it is the boarder of major revision and rejection. Author should revise the paper according reviewers comments.

Reviewer#1

The manuscript looks like a descriptive report but not a research paper. No clear problem statement. No significant objectives. No research gap. No deep discussions. No significant findings. Poor English.

Recommendation: Reject

Reviewer#2

The manuscript introduces a study evaluating the impact of speed humps with raised stripes in Addis Ababa on pedestrian safety. The research employs descriptive and inferential statistical methods, collecting and analyzing accident data from July 2017 to July 2018. It utilizes ANOVA (Analysis of Variance) and paired sample tests to assess the data. The findings indicate that speed humps positively contribute to reducing vehicle speed and minimizing pedestrian accidents, particularly those involving pedestrians crossing the street. While the study has engineering significance, several points warrant further consideration:

1.The "2. Materials and Methods" section of the manuscript seems to only briefly mention the collection of field speed measurements and accident data. It should provide a detailed description of the speed measurement process and the organization and screening of accident data. Furthermore, the subsequent data analysis methods should be validated to ensure that the most appropriate statistical tests were used. The authors may have a misunderstanding regarding statistical significance, and a detailed explanation of the P-values and confidence intervals in the analysis results is necessary.

2. The study appears to have only observed the short-term effects after the intervention, without providing long-term trend data (the statistical data range should be at least 12 months or more). It is recommended to discuss the potential degradation of speed humps over the long term and its impact on effectiveness. It is well-known that the influence of speed humps on safety may change over time.

3. The manuscript does not sufficiently discuss the potential impacts of speed humps or other variables that may affect pedestrian safety. These variables may include, but are not limited to: enforcement of traffic regulations, driver behavior, and road design.

4. The authors mention in the manuscript that the effectiveness of reducing the speed of heavy vehicles and minimizing accidents involving pedestrians walking in a straight line is limited. When considering the impact of speed humps on pedestrians walking in a straight line, one should not overly rely on the quantitative data analysis results but also take into account subjective insights from individuals (qualitative analysis) to gain a more comprehensive understanding.

In summary, the authors must reorganize the article's logic according to scientific writing norms and check all details of the manuscript. If the authors consider continuing to submit this paper for publication in this journal, they should revise and resubmit after thorough verification.

Recommendation: Major revision

Reviewers' comments:

Reviewer's Responses to Questions

**Comments to the Author**

1. Is the manuscript technically sound, and do the data support the conclusions?

Reviewer #1: No

Reviewer #2: Partly

2. Has the statistical analysis been performed appropriately and rigorously? 

Reviewer #1: Yes

Reviewer #2: Yes

3. Have the authors made all data underlying the findings in their manuscript fully available?

Reviewer #1: Yes

Reviewer #2: No

4. Is the manuscript presented in an intelligible fashion and written in standard English?

Reviewer #1: No

Reviewer #2: Yes

5. Review Comments to the Author

Reviewer #1: The manuscript looks like a descriptive report but not a research paper. No clear problem statement. No significant objectives. No research gap. No deep discussions. No significant findings. Poor English.

Reviewer #2: The manuscript introduces a study evaluating the impact of speed humps with raised stripes in Addis Ababa on pedestrian safety. The research employs descriptive and inferential statistical methods, collecting and analyzing accident data from July 2017 to July 2018. It utilizes ANOVA (Analysis of Variance) and paired sample tests to assess the data. The findings indicate that speed humps positively contribute to reducing vehicle speed and minimizing pedestrian accidents, particularly those involving pedestrians crossing the street. While the study has engineering significance, several points warrant further consideration:

1.The "2. Materials and Methods" section of the manuscript seems to only briefly mention the collection of field speed measurements and accident data. It should provide a detailed description of the speed measurement process and the organization and screening of accident data. Furthermore, the subsequent data analysis methods should be validated to ensure that the most appropriate statistical tests were used. The authors may have a misunderstanding regarding statistical significance, and a detailed explanation of the P-values and confidence intervals in the analysis results is necessary.

2.The study appears to have only observed the short-term effects after the intervention, without providing long-term trend data (the statistical data range should be at least 12 months or more). It is recommended to discuss the potential degradation of speed humps over the long term and its impact on effectiveness. It is well-known that the influence of speed humps on safety may change over time.

3.The manuscript does not sufficiently discuss the potential impacts of speed humps or other variables that may affect pedestrian safety. These variables may include, but are not limited to: enforcement of traffic regulations, driver behavior, and road design.

4.The authors mention in the manuscript that the effectiveness of reducing the speed of heavy vehicles and minimizing accidents involving pedestrians walking in a straight line is limited. When considering the impact of speed humps on pedestrians walking in a straight line, one should not overly rely on the quantitative data analysis results but also take into account subjective insights from individuals (qualitative analysis) to gain a more comprehensive understanding.

In summary, the authors must reorganize the article's logic according to scientific writing norms and check all details of the manuscript. If the authors consider continuing to submit this paper for publication in this journal, they should revise and resubmit after thorough verification.

6. PLOS authors have the option to publish the peer review history of their article (what does this mean? ). If published, this will include your full peer review and any attached files.

**Do you want your identity to be public for this peer review?** For information about this choice, including consent withdrawal, please see our Privacy Policy .

Reviewer #1: **Yes: ** Prof. Dr. Ahmed Mancy Mosa

Reviewer #2: No

---

## [Author Response · Author response to Decision Letter 0]

23 Dec 2024

Editor’s Comments

Comment 1: A rebuttal letter that responds to each point raised by the academic editor and reviewer(s). You should upload this letter as a separate file labeled 'Response to Reviewers'.

Response: Thank you very much. We have prepared a rebuttal letter to the academic editor and reviewers.

Comment 2: A marked-up copy of your manuscript that highlights changes made to the original version. You should upload this as a separate file labeled 'Revised Manuscript with Track Changes'.

Response: Thank you. we have marked-up the copy of your manuscript that highlights changes made to the original version

Comment 3: An unmarked version of your revised paper without tracked changes. You should upload this as a separate file labeled 'Manuscript'.

Response: Thank you. We did the same.

Journal Requirements:

Comment 1.Please ensure that your manuscript meets PLOS ONE's style requirements, including those for file naming. The PLOS ONE style templates can be found at

Response: Thank you. We have ensured that our manuscript meets PLOS ONE's style requirements, including those for file naming.

Comment 2. In your Methods section, please provide additional information regarding the permits you obtained for the work. Please ensure you have included the full name of the authority that approved the field site access and, if no permits were required, a brief statement explaining why.

Response: Thank you. We have included that the following information in the methods section.

“Addis Ababa University officially requested permission for the collection of data essential to our study from the city's traffic management agency. The request was done through a formal letter that defined the scope of the study and the data collection requirements. This made it easier to obtain crash and speed data for the period comprising of three years prior to the intervention (installation of the speed hump and rumble strips) and three years after the intervention. Additionally, permission to perform fieldwork was obtained from the Addis Ababa City Administration Police Commission. All study activities were carried out strictly in accordance with the rules outlined in these permissions.”

Reviewer #1:

The manuscript looks like a descriptive report but not a research paper. No clear problem statement. No significant objectives. No research gap. No deep discussions. No significant findings. Poor English.

Response : Thank you very much constructive comments. The effectiveness of speed humps and rumble strips in lowering vehicle speeds and preventing pedestrian collisions has not been assessed, despite the fact that they were initially installed in a particular location. Furthermore, it is yet unknown how different vehicle types such as trucks, buses, and small cars—reduce speed. Investigating how well speed humps with rumble strips reduce vehicle speeds and pedestrian collisions across various vehicle categories is the main goal.

This study is a first in the sector since no other research has looked at how effective speed humps with rumble strips are in Ethiopia. According to the study, tiny car speeds were considerably lowered by speed humps with rumble strips. Their effect on lowering truck and bus speeds, however, was less noticeable, as evidenced by the accompanying improvement in crash rates. By offering data on the effectiveness of speed control methods in an Ethiopian environment, this study makes a significant contribution to the field of road safety, especially in low- and middle-income nations. Future legislative initiatives and infrastructure development targeted at improving pedestrian safety can benefit from the findings.

Reviewer #2:

The manuscript introduces a study evaluating the impact of speed humps with raised stripes in Addis Ababa on pedestrian safety. The research employs descriptive and inferential statistical methods, collecting and analyzing accident data from July 2017 to July 2018. It utilizes ANOVA (Analysis of Variance) and paired sample tests to assess the data. The findings indicate that speed humps positively contribute to reducing vehicle speed and minimizing pedestrian accidents, particularly those involving pedestrians crossing the street. While the study has engineering significance, several points warrant further consideration:

Comment 1: The "2. Materials and Methods" section of the manuscript seems to only briefly mention the collection of field speed measurements and accident data. It should provide a detailed description of the speed measurement process and the organization and screening of accident data. Furthermore, the subsequent data analysis methods should be validated to ensure that the most appropriate statistical tests were used. The authors may have a misunderstanding regarding statistical significance, and a detailed explanation of the P-values and confidence intervals in the analysis results is necessary.

Response: Thank you very much for your constructive comment. As explained in the Materials and Methods section, a speed gun supplied by the Bloomberg Road Safety Initiative was used to measure the speeds. There is a clear description of how to record speed data both before and after the speed hump. Data on accidents was gathered from the City Traffic Management Agency and the City Police Commission. Three years of crash data before and after the speed hump with rumble strips intervention is thought to be adequate to assess the intervention's effectiveness, based on current understanding and pertinent literature. For such studies to be conducted successfully, the literature usually suggests a minimum of three years of crash data.

The p-value, which reflects the findings of the ANOVA and paired t-test, shows a statistically significant difference in mean crashes and operating speeds. As a result, the null hypothesis is rejected and the alternative hypothesis is accepted.To ascertain statistical significance, the study instead uses the F-value (calculated) and the F-critical value, which is derived from the F-distribution table.

Comment 2. The study appears to have only observed the short-term effects after the intervention, without providing long-term trend data (the statistical data range should be at least 12 months or more). It is recommended to discuss the potential degradation of speed humps over the long term and its impact on effectiveness. It is well-known that the influence of speed humps on safety may change over time.

Response: Thank you. The speed observations were conducted over a short period. Prior to finalizing the observation process, a preliminary survey was undertaken, revealing that most speed studies are based on short-duration observations involving approximately 100 vehicles per site. In contrast, crash data were collected for the period preceding the intervention and for three years following the intervention.

Although the number of crashes at a site may fluctuate randomly from year to year, it tends to stabilize around the long-term average over time. This consideration underpins the decision to use three years of crash data for each case, as it provides a robust basis for analyzing the intervention's impact. However, we totally accept the comment on the degradation of long term effects and its impact on effectiveness of speed hump. We have included in the result and discussion section.

Comment 3.The manuscript does not sufficiently discuss the potential impacts of speed humps or other variables that may affect pedestrian safety. These variables may include, but are not limited to: enforcement of traffic regulations, driver behavior, and road design.

Responses: Thank you very much for the valuable comments. The underlying premise of this study is that during the study period, external conditions stayed unchanged. In particular, the road parts that were observed between 2014 and 2017and 2018 and 2021 were not rebuilt. Furthermore, no changes to traffic laws were made throughout this period that would have not affected the study's findings. The findings are unaffected by the August 2024 publication of a new traffic law, which is notable but happened after the study period.

It is recognized that one possible drawback of this study is driver behaviour, which may change over time. The analysis has taken this element into account and included it as a weakness.

Comment 4.The authors mention in the manuscript that the effectiveness of reducing the speed of heavy vehicles and minimizing accidents involving pedestrians walking in a straight line is limited. When considering the impact of speed humps on pedestrians walking in a straight line, one should not overly rely on the quantitative data analysis results but also take into account subjective insights from individuals (qualitative analysis) to gain a more comprehensive understanding.

Response: Thank you very much. We found that, in comparison to smaller vehicles, speed bumps have less of an effect on the speeds of buses and other large vehicles. As you pointed out in your remarks, speed bumps with rumble strips lose some of their effectiveness over time, which lessens their influence on crash-equivalent numbers and vehicle speeds.

Furthermore, buses and big vehicles may pass over speed bumps with comparatively little disturbance to their speed because their clearance height is much higher than that of smaller vehicles. This element plays a part in the intervention's decreased efficacy for various vehicles.

We acknowledge that our previous reliance on quantitative data analysis, while valuable, may have inadvertently overlooked the insights provided by qualitative approaches. This imbalance could potentially lead to conclusions that are not fully reflective of real-world dynamics. To address this limitation, we have incorporated qualitative analysis in this study to gain a more comprehensive and nuanced understanding of the research problem.

In summary, the authors must reorganize the article's logic according to scientific writing norms and check all details of the manuscript. If the authors consider continuing to submit this paper for publication in this journal, they should revise and resubmit after thorough verification.

Response: Thank you very much and we did the same.

---

## [Decision Letter · Decision Letter 1]

10 Jan 2025

Evaluation of the Effectiveness of Speed humps with Rumble Strips on Pedestrian Safety in Addis Ababa, Ethiopia

PONE-D-24-38618R1

Dear Dr. Tulu,

We’re pleased to inform you that your manuscript has been judged scientifically suitable for publication and will be formally accepted for publication once it meets all outstanding technical requirements.

Kind regards,

Mitiku Badasa Moisa

Academic Editor

PLOS ONE

Additional Editor Comments (optional):

I am pleased to inform you that your paper is accepted for publication

Reviewers' comments:

Reviewer's Responses to Questions

**Comments to the Author**

1. If the authors have adequately addressed your comments raised in a previous round of review and you feel that this manuscript is now acceptable for publication, you may indicate that here to bypass the “Comments to the Author” section, enter your conflict of interest statement in the “Confidential to Editor” section, and submit your "Accept" recommendation.

Reviewer #2: All comments have been addressed

2. Is the manuscript technically sound, and do the data support the conclusions?

Reviewer #2: Yes

3. Has the statistical analysis been performed appropriately and rigorously? 

Reviewer #2: Yes

4. Have the authors made all data underlying the findings in their manuscript fully available?

Reviewer #2: Yes

5. Is the manuscript presented in an intelligible fashion and written in standard English?

Reviewer #2: Yes

6. Review Comments to the Author

Reviewer #2: The author has addressed all of my questions, and I now agree to the publication of this article in the esteemed PLOS ONE journal.

7. PLOS authors have the option to publish the peer review history of their article (what does this mean? ). If published, this will include your full peer review and any attached files.

**Do you want your identity to be public for this peer review?** For information about this choice, including consent withdrawal, please see our Privacy Policy .

Reviewer #2: No

---

## [Editor Report · Acceptance letter]

PONE-D-24-38618R1

PLOS ONE

Dear Dr. Tulu,

I'm pleased to inform you that your manuscript has been deemed suitable for publication in PLOS ONE. Congratulations! Your manuscript is now being handed over to our production team.

Kind regards,

on behalf of

Dr. Mitiku Badasa Moisa

Academic Editor

PLOS ONE